# Ethylene Dimerization Performance of NiBTCs Synthesized Using Different Solvents

**Cong Wang, Gang Li *** and **Hongchen Guo**

State Key Laboratory of Fine Chemicals, School of Chemical Engineering, Dalian University of Technology, Dalian 116024, China

* Correspondence: liganghg@dlut.edu.cn

**Abstract:** MOFs have attracted widespread attention in the field of catalytic ethylene dimerization. Compared with post-synthetic modification, ion exchange and other methods to introduce external active centers, the direct use of MOF materials as catalysts is still the most convenient and prospective. Herein, the NiBTCs are synthesized using a one-pot method in two kinds of solvent and characterized by XRD, FT–IR, ICP–OES, XPS, TGA and $N_2$ physical adsorption. After treatment at 150 °C, the catalytic activities of both materials in ethylene dimerization are up to 470.9, and 647.0 $g_{pro.}/(g_{cat.}\cdot h)$ and the selectivity of 1-butene in all products could reach 83.2% and 81.7%, respectively. Stability testing of the catalysts demonstrated that they do not decompose during the reaction, but their reuse performance is degraded. In addition, a probable Cossee–Arlman–type mechanism is proposed. The NiBTCs are shown to have superior catalytic performance in ethylene dimerization compared to employing $Ni(pyz)_2Cl_2$ or $\alpha$–$Ni(im)_2$ as catalysts.

**Keywords:** ethylene dimerization; NiBTCs; MOFs; 1-butene





## 1. Introduction

To benefit from increasing ethylene production, ethylene dimerization has been the focus of researchers' attention since synthesising important industrial feedstocks such as sec-butanol, butadiene and isoprene require large amounts of 1-butene [1–3]. Compared with other metal catalysts, the most widespread process for ethylene dimerization relies on Ni, attributed to its low cost and unique reactivity with unsaturated organics [4–6]. For instance, Chen et al., used $NiCl_2\{(6E,13E)$-$N^1,N^4$-bis(1-(pyridin-2-yl)ethylidene)benzene-1,4-diamine$\}NiCl_2$ as the catalyst in the presence of diethylaluminum chloride ($Et_2AlCl$), the activity was up to 869.1 $g_{pro.}/(g_{cat.}\cdot h)$ at 20 bar, 50 °C [7]; Feng et al., utilized syn-$Ni_2(C_{52}H_{38}Br_4N_2Ni_2P_2)$/EtAlCl$_2$, the activity of reaction could reach 15,307 $g_{pro.}/(g_{cat.}\cdot h)$ [8]. The above homogeneous studies demonstrated the high reactivity of nickel as the active center, but the homogeneous catalytic system still suffers from a complex synthesis process and difficult catalyst recovery, which limits further developments. Heterogeneous catalytic systems can effectively solve the above problems, but improving the reactivity and product selectivity, especially for 1-butene, is still an important challenge to be faced [9].

Metal–organic frameworks (MOFs) are solid-phase porous materials composed of organic ligands and metal clusters [10–14]. Similar to homogeneous catalysts, they possess rich coordination patterns and diverse organic ligand structures, which provide a feasible idea for efficient catalytic ethylene dimerization in heterogeneous phases [15]. The studies of Ni-MOFs catalysts showed that the coordination of unsaturated sites (CUS) in the materials could effectively improve the catalytic activity in ethylene dimerization [16–18]. CUS-containing MOFs are currently prepared in the following ways: (1) Post-synthetic modification to introduce a new active center, e.g., Canivet et al., grafted $Ni(2–PyCHO)Cl_2$ in $NH_2$–MIL(Fe)–101 to form Ni@(Fe)MIL–101 for ethylene dimerization with a maximum

activity of 205.0 g$_{pro}$./(g$_{cat}$.·*h*) [19]. (2) Creation of active centers by ion exchange, e.g., Metzger et al., replaced a portion of Zn in MFU-4l with Ni by ion exchange, resulting in the formation of CUS. The activity was up to 198.0 g$_{pro}$./(g$_{cat}$.·*h*) at 50 bar, 25 °C, and the selectivity of 1-butene in the product reached 92.0% [20]. (3) Creation of active centers by inconsistent coordination mode, e.g., Chen et al., used a one-pot method to add a certain amount of Ni to the synthesis of ZIF-8 to form the CUS. The reactivity reached 2130 g$_{pro}$./(g$_{cat}$.·*h*) at 35 °C, 30 bar, and the selectivity of 1-butene in the product reached 85.1% [21]. (4) Removal of solvent molecules ligated to the active center in the structure of the MOF, e.g., Hu et al., reported that Ni-UMOFNs require activation at 190 °C in a vacuum oven to achieve better catalytic activity and 1-butene selectivity (5536 h$^{-1}$ and 75.6%) [17]. Comparing these methods indicates that the fourth way is the most convenient and has relatively better prospects for industrial applications, which is worth exploring further.

NiBTC was firstly synthesized from nickel(II) acetate tetrahydrate and 1,3,5-benzene tricarboxylic acid (H$_3$BTC) through a classical hydrothermal reaction, which is reported by O. M. Yaghi and coworkers [22]. Subsequently, NiBTCs with different coordination modes have been synthesized by changing the synthesis conditions (e.g., reaction temperature, reaction time, solvent type and additives required for crystallization), which have been widely used, such as OER, supercapacitor, anode material for Li-ion batteries and catalysts for the reaction of CO$_2$ cycloaddition [23–25]. In general, a benzene tricarboxylic acid ion is coordinated to four nickel atoms with two multidentate carboxylate groups in the structure of NiBTCs. In order to complete the octahedral environment of the nickel center, the remaining two empty coordination sites are occupied by water molecules (or solvent molecules) (Figure 1), which can be dispelled to form CUS active centers [26]. This is the key to how NiBTC can be used in the aforementioned research, so the reactivity of NiBTC materials in ethylene dimerization is expected.

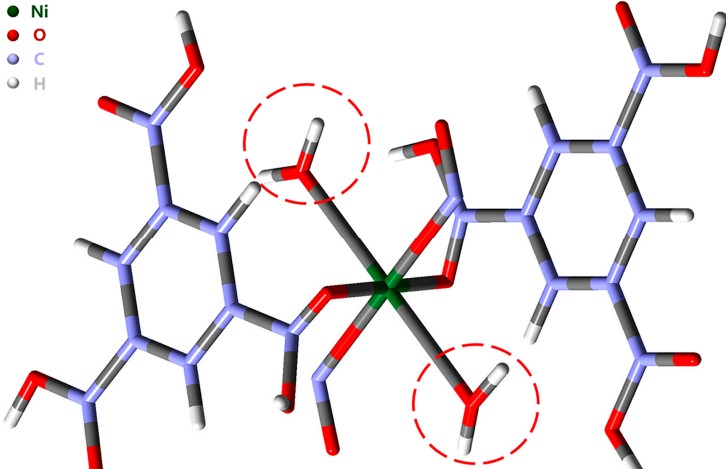

**Figure 1.** Probable tube view of nickel atoms in their environment with NiBTCs (Red circles indicated coordination water molecules) [26,27].

In this work, we synthesized two NiBTCs using a one-pot method in different solvents and applied them as catalysts in the dimerization of ethylene. The activities of the NiBTCs were up to 470.9, and 647.0 g$_{pro}$./(g$_{cat}$.·*h*), and the selectivity of 1-butene in all products could reach 83.2% and 81.7%, respectively. The fresh and used catalysts were also characterized by inductively coupled plasma optical emission spectrometer (ICP–OES), X-ray diffraction (XRD), Fourier transform infrared spectroscopy (FT–IR), X-ray photoelectron spectroscopy (XPS), thermogravimetric analysis (TGA) and N$_2$ physical adsorption. Two other MOFs (Ni(pyz)$_2$Cl$_2$ and α–Ni(im)$_2$) were also synthesized to compare the catalytic activity with the NiBTCs. Moreover, a possible ethylene dimerization mechanism of NiBTC was proposed with the inspiration of the Cossee–Arlman mechanism.

## 2. Results and Discussion

### 2.1. Characterization

The synthesis process of the two NiBTCs is described in Section 3.2. XRD patterns are used to investigate the structure of the NiBTCs. As shown in Figure 2A, the NiBTC–DMF presents two distinct diffraction peaks at 11° and 23° corresponding to the (100) and (101) planes, respectively, which broaden in NiBTC–EtOH. The phenomenon observed in this work is similar to the results previously reported in NiBTC synthesis [23,24]. The FT–IR spectra clearly show peaks for the NiBTC–DMF and NiBTC–EtOH in Figure 2B. The wide peak centered at 3408 cm$^{-1}$ is attributed to the stretching mode of hydroxyl (O–H) groups from absorbed water, and the bands in the region of 780–722 cm$^{-1}$ correspond to the in-plane as well as the out-of-plane deformation vibrations of the C-H groups in the benzene ring [28]. Meanwhile, by magnifying the part of 1800–1200 cm$^{-1}$ (Figure 2C), it can be seen that the adsorption bands of -COOH, which relate to the BTC ligands and formic ligands, are not observed in the expected region at 1800–1680 cm$^{-1}$ [22,29]. On the other hand, the characteristic bands of coordinated -COO groups located at 1618–1559 cm$^{-1}$ (asymmetric vibration) and 1436–1307 cm$^{-1}$ (symmetric vibration) are detected in the corresponding spectral region [30,31]. All of the above implies that the BTC and formic ligands have coordinated with the metal ions [32]. It is also noteworthy that two distinct absorption peaks are present in the structure of NiBTC–DMF alone; the peaks at 2168 cm$^{-1}$ and 1113 cm$^{-1}$ should be attributed to the tertiary amine group and the stretching vibration of C-N, respectively [24,33]. These two absorption peaks also prove the presence of the solvent DMF molecule in the structure of NiBTC–DMF.

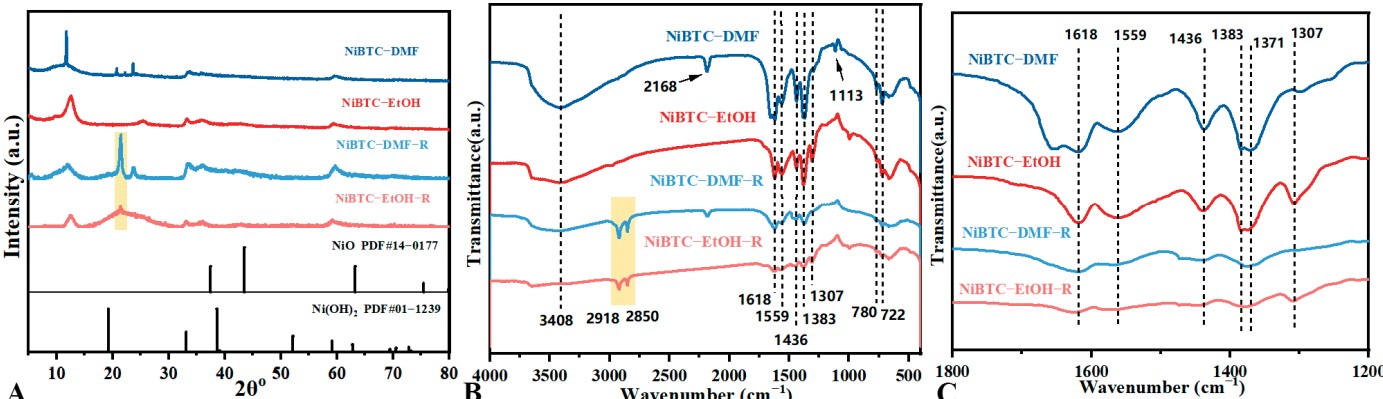

**Figure 2.** (**A**) XRD patterns of NiBTC–DMF (blue) and NiBTC–EtOH (red); (**B**) FT-IR spectra of NiBTC–DMF (blue) and NiBTC–EtOH (red). (**C**) FT-IR spectra of NiBTC–DMF (blue) and NiBTC–EtOH (red) at 1800–1200 cm$^{-1}$.

XPS characterization was carried out, and the corresponding results are presented in Figure 3. The survey spectra of the NiBTCs (Figure 3A) exhibit three obvious peaks at 285 (C 1 s), 531 (O 1 s) and 856 (Ni 2p3), suggesting the formation of MOFs. For the Ni 2p core level XPS spectra, as shown in Figure 3B, two major peaks positioned at 855.6 and 873.1 eV can be assigned to the splitting of Ni 2p 3/2 and Ni 2p 1/2, a spin–energy separation of 17.5 eV could be calculated from the major peaks, also confirming the presence of Ni$^{2+}$ [34–36]. The deconvoluted higher binding energy peaks at 857.0 and 874.5 eV may be attributed to the trace amount of Ni$^{3+}$, which may be generated by the hydrothermal and oxidative effects during the crystallization process [37,38]. The broad peaks located at 861.7 eV and 879.7 eV are the satellite peaks of Ni 2p 3/2 and Ni 2p 1/2. Furthermore, Figure 3C shows the spectra of C 1 s, with the three main peaks at 284.8 eV (C–C), 285.6 eV (C–O) and 288.2 eV (C=O) nearby, which is in good agreement with the bonding mode of element C in NiBTC. Figure 3D displays the spectra of O 1 s, and the O peak at 531.2 eV is typical of Ni–oxygen (Ni-O) bonding and the deconvoluted binding energy peaks at 533.0 eV may be attributed to C-O bonding [39].

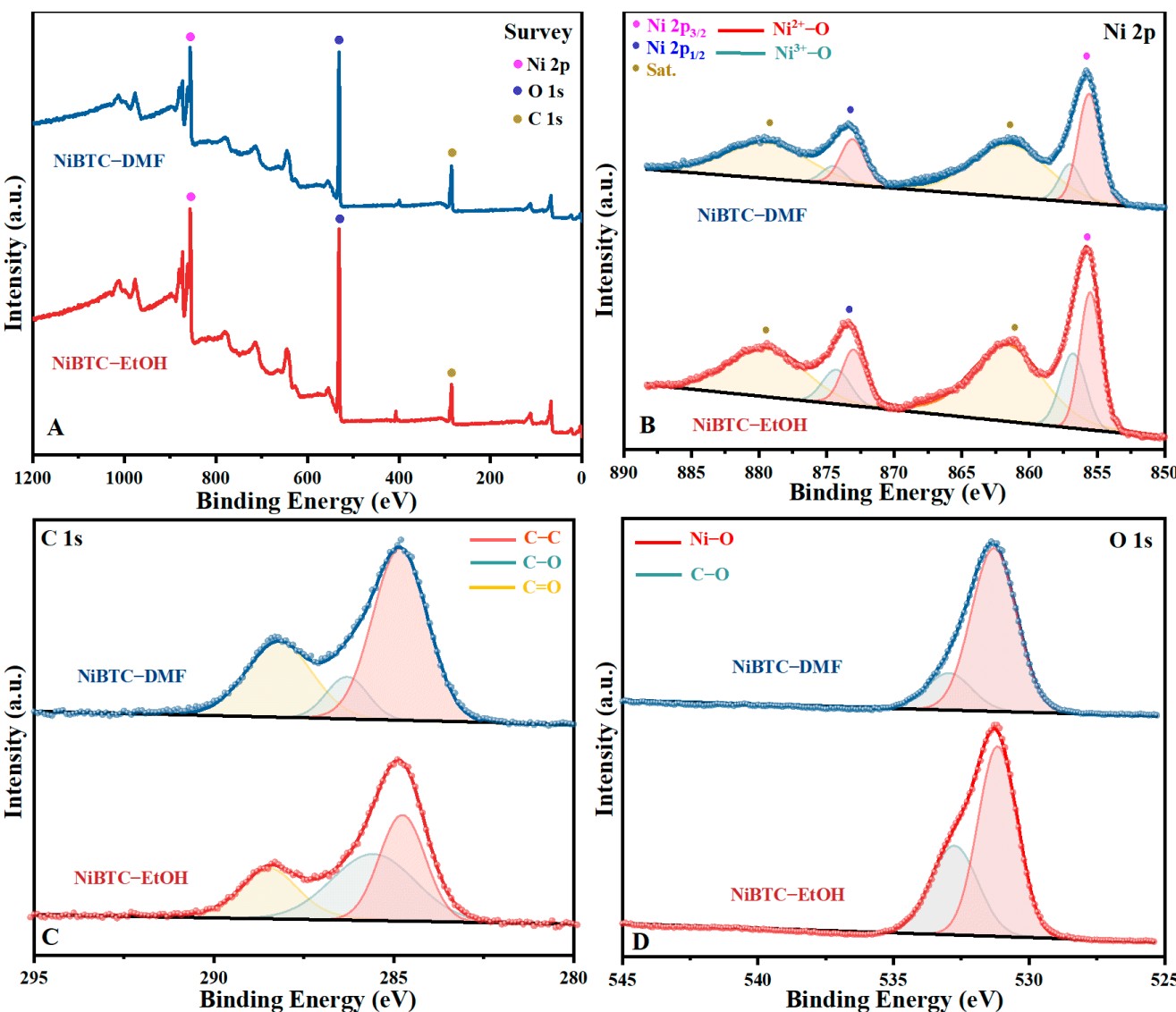

**Figure 3.** XPS patterns of NiBTC–DMF (blue) and NiBTC–EtOH (red): (**A**) survey, (**B**) Ni 2p, (**C**) C 1 s and (**D**) O 1 s.

Thermogravimetric analyses at 40–600 °C are performed under an $N_2$ atmosphere, and the results are shown in Figure 4A,B. Each thermogram of the NiBTCs has three distinct regions at different temperature ranges. Their initial weight loss steps (NiBTC–DMF, 40–178 °C and NiBTC–EtOH, 40–229 °C) are attributed to the loss of water molecules and some of the solvent molecules in coordination [40,41]. As the heating continued, their structures became unstable. The weight losses of 16.7 and 17.3% in the same sequence are attributed to the further separation of solvent molecules (DMF, EtOH) from the structure until 350 and 361 °C, respectively [24,42]. The network of NiBTCs collapses completely when the temperature is higher than 450 °C [43]. The results above further prove that the water molecules exist in the NiBTCs, which is in accordance with the FT–IR, and also demonstrate that two NiBTCs can get rid of $H_2O$ and some solvent molecules by heat treatment below 178 and 229 °C, respectively, to form CUS.

The NiBTCs present a typical type–IV isotherm with an H4–type hysteresis loop starting from $P/P_0 \approx 0.45$, indicating the presence of interparticle buildup (Figure 4C). The specific surface area of NiBTC–DMF (60.91 $m^2$/g) is somewhat higher than the values reported previously (40.36 $m^2$/g) [24]. This is primarily because the material was subjected to the activation process following synthesis, which is commonly performed to achieve

a high surface area for the MOFs [27]. In addition, the micropore area and volume of the NiBTCs are calculated to be approximately equal to 0 by the t-plot method, which is consistent with the same type of NiBTCs [24,44–46]. This proves that they may not possess the microporous structure possessed by MOFs in general, and thus their specific surface area is lower than that of other structural NiBTCs [42,47].

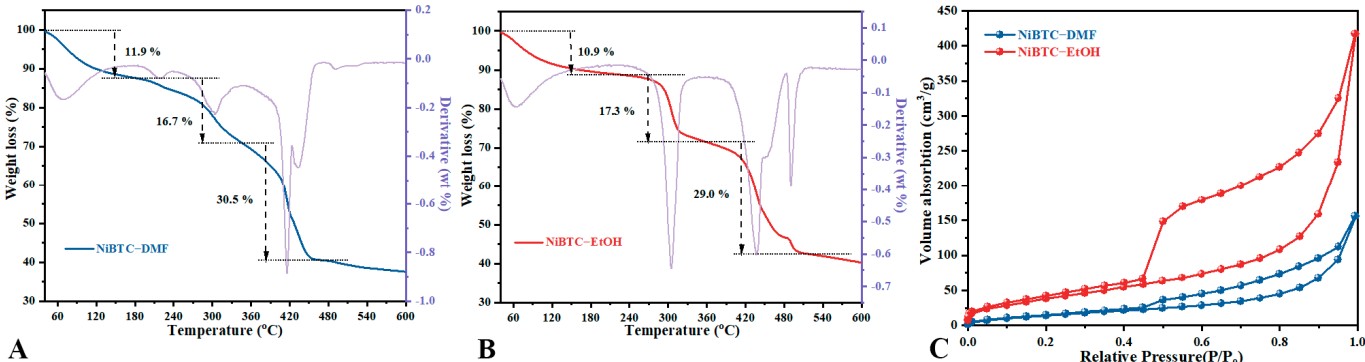

**Figure 4.** TGA curves (**A**,**B**) and $N_2$ adsorption-desorption isotherms of NiBTCs (**C**).

## 2.2. Ethylene Dimerization

The catalytic performance of NiBTC–DMF is shown in Figure 4. The effect of the Al/Ni molar ratio on the catalytic performance of NiBTC–DMF in ethylene dimerization is investigated in Figure 5A. With the increase of the Al/Ni molar ratio from 50 to 200, the catalytic activity first increases and then decreases. The product distribution gradually shifts toward the high carbon number with an increase of $Et_2AlCl$. However, the obtained $\geq C_8$ products are not linear $\alpha$-olefins but branched olefins as well as other impurities, so it can be basically speculated that the addition of excess $Et_2AlCl$ may interfere with the formation of active species and lead to the over-reduction of the active center, which is consistent with the reports in the literature [48]. The effects of different temperatures on the catalytic activity of NiBTC–DMF are shown in Figure 5B. The reactivity rises gradually with the increase of temperature from 20 to 50 °C, reaching a maximum of 196.9 $g_{pro.}/(g_{cat.} \cdot h)$ at 50 °C. After that, the reactivity shows a large decrease with the increase in temperature. The dimerization is exothermic, and increasing the temperature will further lead to a decrease in the solubility of ethylene in the solvent, thus leading to a decrease in activity [49]. The activity decreases from 196.9 $g_{pro.}/(g_{cat.} \cdot h)$ to 153.0 $g_{pro.}/(g_{cat.} \cdot h)$ when the amount of catalyst is reduced from 0.005 g to 0.003 g (Figure 5C), indicating that the number of active centers of the catalyst also affects the dimerization reaction. Therefore, it concluded that the best performance of the catalyst could be obtained at 0.005 g, Al/Ni molar ratio of 100 and a temperature of 50 °C based on the above.

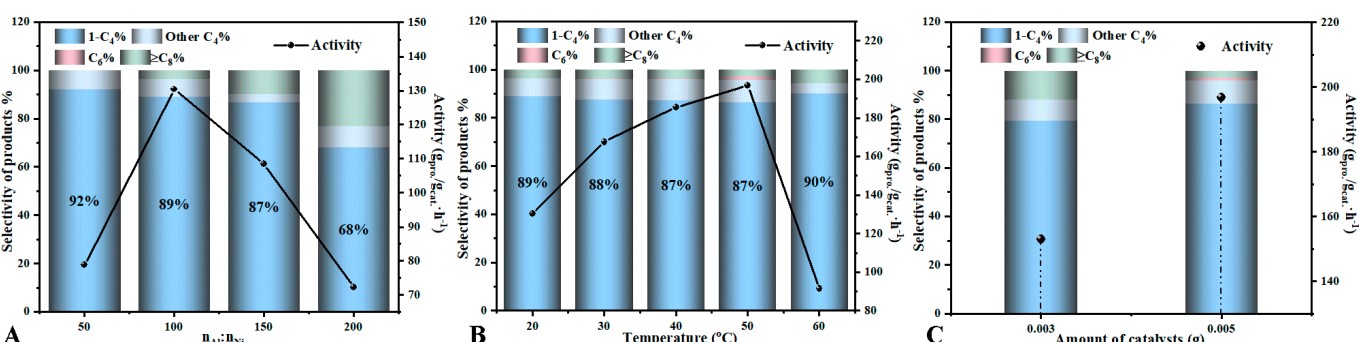

**Figure 5.** (**A**) Effect of Al/Ni; (**B**) Effect of temperature and (**C**) Effect of the amounts of catalyst on the catalytic activity and the product distribution. Reaction conditions: Ethylene pressure of 10 bar, 0.5 h, 400 rpm, NiBTC–DMF activated by 150 °C and $Et_2AlCl$ as a co-catalyst.

Under these optimal conditions, we compare the activity of the NiBTC–DMF and NiBTC–EtOH as shown in Table 1. There is no product only in the presence of Ni(NO$_3$)$_2$·6H$_2$O, H$_3$BTC and Et$_2$AlCl or only in the presence of H$_3$BTC and Et$_2$AlCl, which indicates that it is the MOFs instead of metal source or organic ligands that possess the catalytic activity in the reaction. The activity of NiBTC–EtOH is always higher than NiBTC–DMF's, but the selectivity of 1-butene has the opposite trend. The reason may be that the active center in NiBTC–EtOH with low crystallinity is more fully exposed, so its activity is higher, but the effect of the surrounding ligands on the β-H elimination is relatively weakened at the same time. As a result, 1-butene stays in the active center, and further branching reactions occur. After extending the reaction time, we find that the activity of both catalysts decreases, indicating that many products may block or cover the active center. Thus, NiBTC–EtOH, with more active centers, exhibits higher reactivity (279.4 > 157.2 g$_{pro.}$/g$_{cat.}$·$h^{-1}$) in the reaction. The TOF value of the catalysts is calculated based on the Ni content in Section 3.2.

**Table 1.** Ethylene dimerization with NiBTCs.

| Catalyst | Time (h) | Ethylene Pressure (bar) | Activity g$_{pro.}$/(g$_{cat.}$·h) | TOF ($h^{-1}$) | Selectivity (%) | | | |
| --- | --- | --- | --- | --- | --- | --- | --- | --- |
| | | | | | 1-C$_4$ | Other C$_4$ | C$_6$ | ≥C$_8$ |
| Ni(NO$_3$)$_2$ 6H$_2$O + H$_3$BTC + Et$_2$AlCl | 0.5 | 10 | 0 | 0 | 0 | 0 | 0 | 0 |
| H$_3$BTC + Et$_2$AlCl | 0.5 | 10 | 0 | 0 | 0 | 0 | 0 | 0 |
| NiBTC–DMF | 0.5 | 10 | 196.9 | 1325 | 86.5 | 9.40 | 1.30 | 2.80 |
| | 1.0 | 10 | 157.2 | 1057 | 76.9 | 16.5 | 2.30 | 4.30 |
| | 0.5 | 20 | 470.9 | 3167 | 83.2 | 8.30 | 2.51 | 5.99 |
| NiBTC–EtOH | 0.5 | 10 | 291.6 | 1764 | 78.0 | 16.9 | 1.55 | 3.55 |
| | 1.0 | 10 | 279.4 | 1691 | 77.9 | 14.7 | 2.87 | 4.53 |
| | 0.5 | 20 | 647.0 | 3915 | 81.7 | 12.3 | 2.76 | 3.24 |

Reaction conditions: 50 °C, 400 rpm, 0.005 g NiBTCs activated by 150 °C, Al:Ni = 100, Et$_2$AlCl as a co-catalyst.

### 2.3. The Stability of Catalyst

To verify the stability of the catalyst in solution, we separate the filtrate of the reaction under anhydrous and oxygen-free conditions and put the filtrate back into the reaction. The result of the leaching test shows that the reaction does not proceed any further, even in the presence of an Al-based co-catalyst (Table 2), which demonstrates that the catalytic system is always a heterogeneous system.

**Table 2.** Comparison of the amount of product obtained from the leaching test.

| Entry | Amount of Product Obtained (g) | Amount of Product Obtained from the Filtrate (g) |
| --- | --- | --- |
| NiBTC–DMF | 0.492 | 0.009 |
| NiBTC–EtOH | 0.729 | 0.035 |

Leaching test conditions: Ethylene pressure of 10 bar, 400 rpm, NiBTC activated by 150 °C and Et$_2$AlCl as a co-catalyst, 0.5 h.

The recovered catalysts are fully washed with anhydrous ethanol, dried and further tested after being reactivated. The results are shown in Table 3. The activity of both materials decreases, with NiBTC–DMF decreasing to 54% of the original activity and NiBTC–EtOH decreasing to 41%. In order to determine the factors involved, the catalysts are characterized by ICP–OES, XRD and FT–IR after the first reaction. It can be seen that the percentage of Al in the spent catalyst increases significantly, indicating that some of the Et$_2$AlCl hydrolysis products are attached to the catalyst [21]. The XRD patterns of the recovered catalysts show a significant decrease in the crystallinity, while the shape of the original peaks remains essentially unchanged in Figure 2A. Visibly, new diffraction peaks appear at 21.5°, indicating a change in the crystal structure of both materials. FT–IR spectra show a decrease in the intensity but no change in the position of the absorption peaks

for both materials in Figure 2B and C. Additional absorption peaks appear at 2918 cm$^{-1}$ and 2850 cm$^{-1}$, which are attributed to the stretching vibration peaks of alkanes. All of the above indicates that Al-impurities cover the active center, and some alkyl groups are embedded in the skeletons of both materials, resulting in decreased activity of the recovered catalysts.

**Table 3.** Ethylene dimerization with recovered NiBTCs.

| Entry | Ni Content % [a] | Al Content % [a] | Activity $g_{pro.}/(g_{cat.}\cdot h)$ | TOF (h$^{-1}$) | Selectivity (%) | | | |
|---|---|---|---|---|---|---|---|---|
| | | | | | 1-C$_4$ | Other C$_4$ | C$_6$ | $\geq$C$_8$ |
| NiBTC–DMF–R | 20.89 | 7.54 | 106.2 | 714.3 | 85.1 | 4.31 | 0.39 | 10.2 |
| NiBTC–EtOH–R | 24.77 | 6.54 | 121.0 | 732.1 | 74.9 | 11.3 | 1.30 | 12.5 |

Reaction conditions: Ethylene pressure of 10 bar, Al:Ni = 100, 50 °C, 0.5 h, 400 rpm, 0.005 g NiBTC activated by 150 °C and Et$_2$AlCl as a co-catalyst. [a] Measured by ICP-OES at the same concentration (0.001 g/L).

### 2.4. Postulated Mechanism for Ethylene Dimerization

According to the above results, we propose a possible mechanism for the NiBTCs-catalyzed ethylene dimerization, as shown in Figure 6A. At first, NiBTC is activated by the high temperature at 150 °C to eliminate the water/solvent molecules in the structure, followed by further activation by Et$_2$AlCl to form Ni-H active centers. Then, an incoming ethylene is coordinated to the active Ni, followed by the ethylene insertion and producing intermediates Ni-C$_2$H$_5$. After another ethylene subsequently coordinates to Ni-C$_2$H$_5$ and inserts into the C-H bond, the Ni-C$_4$H$_9$ is formed. The β-H elimination of Ni-C$_4$H$_9$ would generate 1-butene as well as release the active center. When the β-H elimination does not occur in time, further branching reactions are carried out [20]. The generation of the C$_6$ products is essentially the same as that of the C$_4$ products through an insertion–elimination process. The distribution of the resulting C$_4$ products (1-butene, cis-2-butene, trans-2-butene and isobutene) and C$_6$ products (3-methyl-1-propene, 1-hexene, 2-ethyl-1-butene, 3-hexene, 2-hexene and benzene) (Figure 6B) are similar to previous studies [50]. Combining with other reported reaction mechanisms of nickel-based catalysts, we suggest that NiBTCs follow the Cossess–Arlman type mechanism for the reaction [51–53].

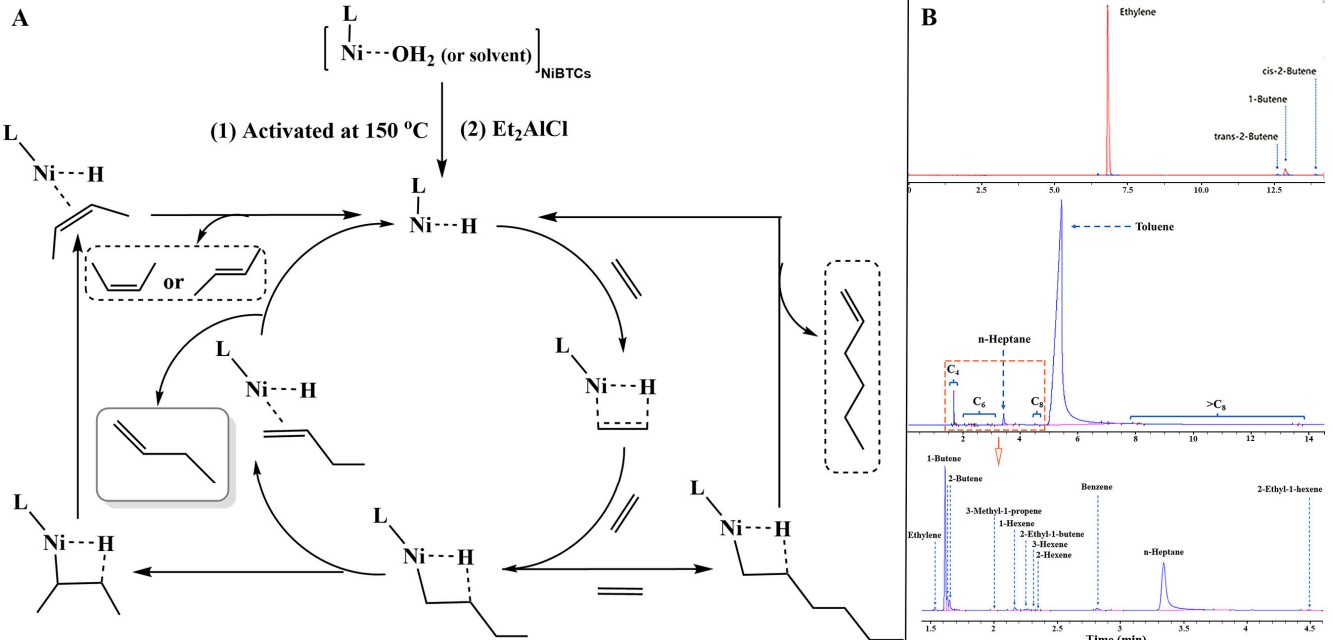

**Figure 6.** (**A**) Postulated catalytic cycle for ethylene dimerization by NiBTCs; (**B**) Chromatogram of the products after ethylene dimerization with NiBTC–EtOH for 1 h.

### 2.5. Comparison with Other Catalysts

In this work, the Ni(pyz)$_2$Cl$_2$ and α–Ni(im)$_2$ are characterized by XRD and FT–IR (Figure 7A,B). It can be seen that the simulated and experimental XRD patterns are in good agreement with each other, which proves the phase purity of the as-synthesized samples [54,55]. In FT–IR spectra, the weak peaks in the range of 3143–3108 cm$^{-1}$ are attributed to the stretching mode of C-H on the heterocyclic compounds and the range of 825–665 cm$^{-1}$ is attributed to the out-of-plane bending vibration of C=N [56]. By magnifying the part of 1800–1200 cm$^{-1}$ (Figure 7C), for Ni(pyz)$_2$Cl$_2$, the band at 1636 cm$^{-1}$ and the range of 1487–1417 cm$^{-1}$ are attributed to the stretching vibrations C=C and C-H groups, respectively. The band at 1390 cm$^{-1}$ is assigned to the stretching vibration of C=N [57]. Unlike Ni(pyz)$_2$Cl$_2$, it can be stated that the stretching vibrations of C=N are superimposed at 1587 cm$^{-1}$ for α–Ni(im)$_2$ [58]. The stretching vibrations of C=C and C-N groups are located in the range of 1689–1619 cm$^{-1}$ and 1489–1470 cm$^{-1}$, respectively [59]. In addition, the peaks at 1321–1233 cm$^{-1}$ correspond to the in-plane bending of the imidazole rings [60].

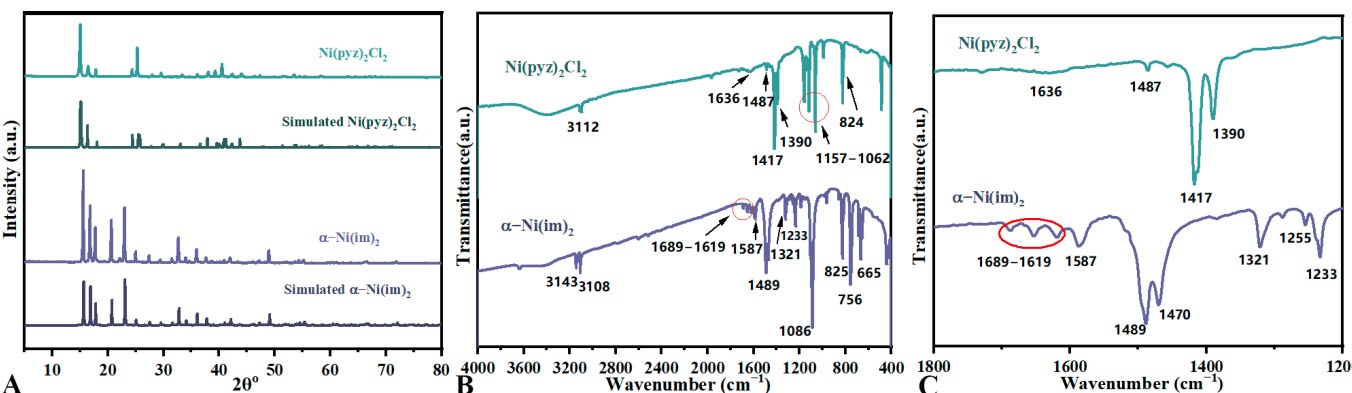

**Figure 7.** (**A**) XRD patterns of Ni(pyz)$_2$Cl$_2$ (green) and α–Ni(im)$_2$ (purple); (**B**) FT–IR spectra of Ni(pyz)$_2$Cl$_2$ (green) and α–Ni(im)$_2$ (purple); (**C**) FT–IR spectra of Ni(pyz)$_2$Cl$_2$ (green) and α–Ni(im)$_2$ (purple) at 1800–1200 cm$^{-1}$.

The catalysts are subjected to ethylene dimerization under the same reaction conditions (Table 4), and the results show that the activity and TOF values of the two NiBTCs are higher than those of the other two materials. The reason may be that the coordination site of Ni(pyz)$_2$Cl$_2$ was occupied by Cl, which needed to be replaced by an active group in the aluminum co-catalyst for further reaction, while α–Ni(im)$_2$ is a four-ligand structure with a large space resistance around the empty coordination site, so their activity is relatively lower.

**Table 4.** Comparison of catalytic performance of Ni-MOFs in ethylene dimerization.

| Entry | Activity g$_{pro.}$/(g$_{cat}$·h) | TOF (h$^{-1}$) | Selectivity (%) | | | |
|---|---|---|---|---|---|---|
| | | | 1-C$_4$ | Other C$_4$ | C$_6$ | ≥C$_8$ |
| NiBTC–DMF | 196.9 | 1325 | 86.5 | 9.40 | 1.30 | 2.80 |
| NiBTC–EtOH | 291.6 | 1764 | 78.0 | 16.9 | 1.55 | 3.55 |
| Ni(pyz)$_2$Cl$_2$ | 149.7 | 1261 | 79.1 | 12.2 | 0.78 | 7.92 |
| α–Ni(im)$_2$ | 184.7 | 1075 | 71.9 | 18.7 | 1.78 | 7.62 |

Reaction conditions: Ethylene pressure of 10 bar, Al:Ni = 100, 50 °C, 0.5 h, 400 rpm, 0.005 g catalyst activated by 150 °C and Et$_2$AlCl as a co-catalyst.

The activities of NiBTCs in ethylene dimerization are also compared with other reported catalysts in Table 5, which indicates that better activity and selectivity of 1-C$_4$ can be achieved using NiBTCs under optimal reaction conditions.

**Table 5.** Comparison of reported catalytic activity of Ni–MOFs for ethylene dimerization.

| Entry | Ethylene Pressure (bar) | Al/Ni Molar Ratio | Activity $g_{pro.}/(g_{cat.} \cdot h)$ | Selectivity of 1-C$_4$ (%) | Ref. |
|---|---|---|---|---|---|
| Ni–ZIF–8 | 50 | 4640 | 2130 | 85.1 | [21] |
| 45 Ni–ZIF–L | 30 | 7187 | 71.96 | 86.3 | [61] |
| Ni@TAPA–CPPs | 5 | 500 | 251.92 | 29.6 | [62] |
| Ni@MAPA–COF | 7 | 500 | 202.51 | 47.5 | [63] |
| Ni@MOF | 10 | 800 | 43.52 | 43.7 | [51] |
| 30Ni@(Fe)–MIL–101 | 30 | 70 | 205.0 | Not report | [19] |
| Ni(1%)–MFU–4l | 50 | 500 | 198.0 | 92.0 | [20] |
| Ni(7.5%)–CFA-1 | 50 | 2000 | 22.23 | 87.1 | [64] |
| IRMOF–3–Ni | 20 | 100 | 125.8 | 35.0 (C$_4$ %) | [65] |
| MixMOF–Ni-b | 20 | 100 | 492.9 | 92.7 (C$_4$ %) | [65] |
| NiBTC–DMF | 20 | 100 | 470.9 | 83.2 | This work |
| NiBTC–EtOH | 20 | 100 | 647.0 | 81.7 | This work |

## 3. Materials and Methods

### 3.1. Reagents and Instruments

Ni(NO$_3$)$_2$·6H$_2$O (98%), Ni(CH$_3$COO)$_2$·4H$_2$O (98%), NiCl$_2$·6H$_2$O (98%), n-heptane (98.5%) and ethanol (99.7%) were purchased from Tianjin Damao Chemical Reagent Factory (Tianjin, China). Imidazole (99%), pyrazine (99%) and Et$_2$AlCl (1.0 M solution in toluene) were purchased from Aladdin Biochemical Technology Co., Ltd. (Shanghai, China). H$_3$BTC (99%) was purchased from J & KCHEMICAL Ltd. (Beijing, China). Toluene (99.8%) and acetone (99.9%) were purchased from Tianjin Kemiou Chemical Reagent Co., Ltd. (Tiangjin, China). N, N-dimethylformamide (DMF; 99%) was purchased from Tianjin Fuyu Fine Chemical Co., Ltd. (Tianjin, China). Deionized water was used in all the experimental processes.

XRD patterns are performed at a scanning speed of 0.15 s/step on a Bruker D8 Advance diffractometer (Bruker, Karlsruhe, Germany) equipped with Cu Kα radiation (λ: 1.5418 Å; voltage: 40 kV; current: 40 mA) in the range of 5–80° with a step of 0.025°.

FT–IR spectra are obtained using a Bruker EQUINOX55 spectrometer (Bruker, Karlsruhe, Germany) from 400 cm$^{-1}$ to 4000 cm$^{-1}$ by the standard KBr disk method.

ICP–OES analysis is obtained using a PerkinElmer Avio 500 plasma emission spectrometer (PerkinElmer, Waltham, MA, USA).

XPS analysis is obtained using a Thermo Scientific K–Alpha+ spectrometer (ThermoFisher, Waltham, MA, USA) with Al Ka radiation (1486.6 eV). The binding energy (B.E.) spectrum was calibrated according to the C 1s standard spectrum (B.E. = 284.8 eV). The composition on the surface of the catalysts according to the atomic ratios was calculated, and the Linear background and Gaussian–Lorentzian methods were used for peak analysis.

TGA curves were obtained by a TGA8000–Frontier–Clarus SQ8T (PerkinElmer, Waltham, MA, USA) from 30 to 600 °C at a ramping rate of 10 °C/min in N$_2$.

The specific surface area of the synthesized NiBTCs was calculated using the Brunauer–Emmett–Teller (BET) equation based on the N$_2$ adsorption–desorption isotherms (collected at 77 K) recorded by a Micromeritics AutoChem II 2920 instrument (Micromeritics, Norcross, GA, USA).

### 3.2. Catalysts Preparation

NiBTC–DMF and NiBTC–EtOH. The procedure to synthesize NiBTC–DMF was similar to previously reported literature with a few modifications [24]. 3.0 mmol of Ni (NO$_3$)$_2$·6H$_2$O (0.8724 g) and 0.5 mmol of H$_3$BTC (0.1050 g) dissolved in 60 mL DMF. Thereafter, the mixed solution was stirred for 1 h at room temperature, then transferred into a 100 mL Teflon-lined autoclave at 150 °C for 12 h. After cooling to room temperature, the precipitate was centrifuged and washed with DMF and ethanol several times. Finally, the obtained product was dried in the oven for 24 h at 80 °C. The synthesis process of NiBTC–EtOH was basically the same as the NiBTC–DMF, except that 60 mL DMF was replaced by 60 mL EtOH. A

schematic illustration of the synthesis procedure and the probable structure for the NiBTCs is shown in Scheme 1.

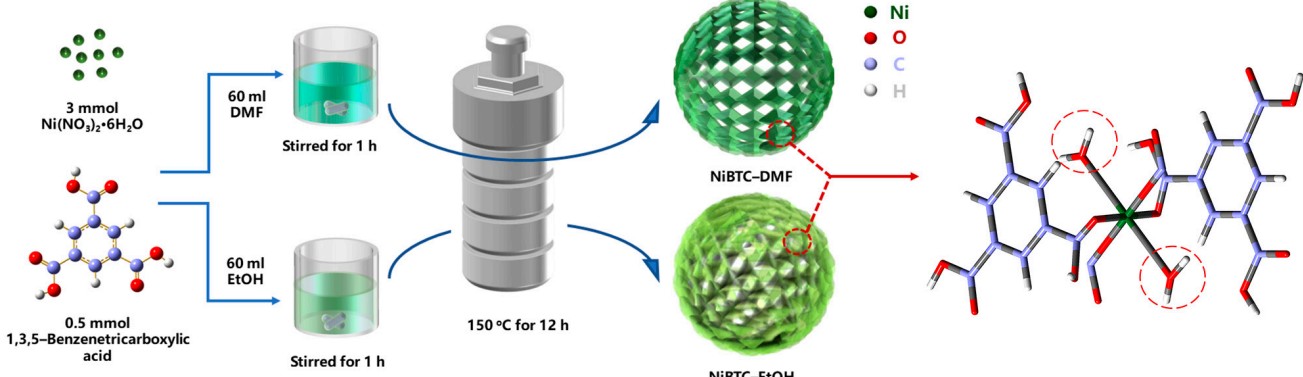

**Scheme 1.** Schematic illustration describing the synthesis of NiBTC–DMF and NiBTC–EtOH.

$Ni(pyz)_2Cl_2$. The procedure to synthesize $Ni(pyz)_2Cl_2$ was similar to previously reported literature with a few modifications [54]. A weight of 0.4 g (5.0 mmol) pyrazine was dissolved in 15 mL of acetone and added slowly dropwise with stirring to 10 mL of a water solution containing 0.594 g (2.5 mmol) of $NiCl_2 \cdot 6H_2O$. After stirring for 1 h, a greenish-white precipitate was obtained by centrifugation, repeatedly washed with water and acetone, and dried under a vacuum. A schematic illustration of the synthesis procedure and the structure for $Ni(pyz)_2Cl_2$ is shown in Scheme 2.

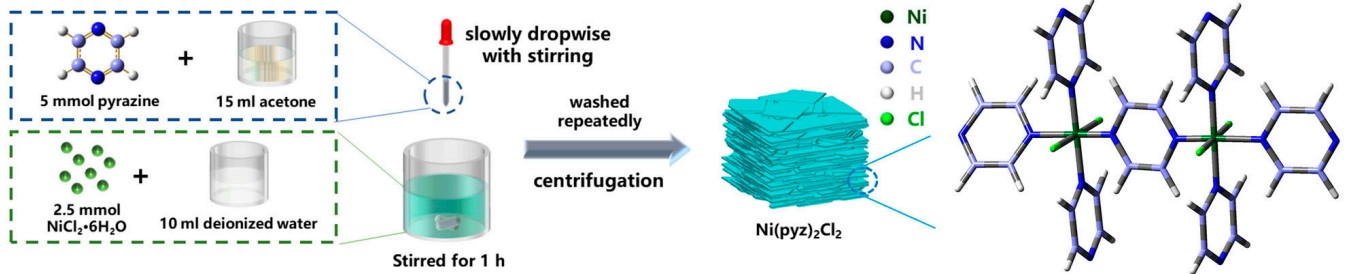

**Scheme 2.** Schematic illustration describing the synthesis of $Ni(pyz)_2Cl_2$.

$\alpha$–$Ni(im)_2$. The procedure to synthesize $\alpha$–$Ni(im)_2$ was similar to the previously reported literature with a few modifications [55]. Imidazole (1.17 g, 17 mmol) was added to 50 mL of aqueous nickel (II) acetate (500 mg, 2.83 mmol) to obtain a dark blue solution. After adding ammonia solution (5 mL) drop by drop, the solution was allowed to stand for about 30 min, heated to 100 °C and stirring was started. After 5 h of reaction, a yellow precipitate was obtained, filtered, washed with deionized water (4 × 30 mL) and acetone (2 × 30 mL), and dried under vacuum at 80 °C for 12 h. A schematic illustration of the synthesis procedure and the structure for $\alpha$–$Ni(im)_2$ is shown in Scheme 3.

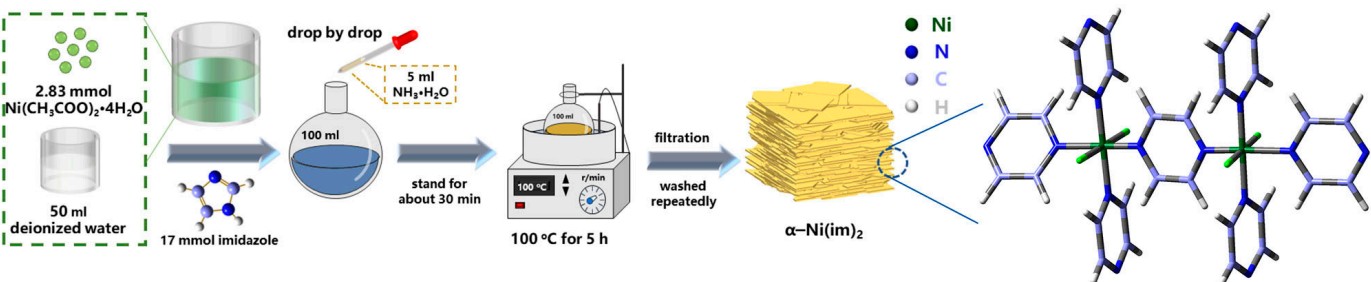

**Scheme 3.** Schematic illustration describing the synthesis of $\alpha$–$Ni(im)_2$.

The yields of all catalysts above and the Ni content are shown in Table 6.

**Table 6.** The yields of the Ni-MOFs and nickel content.

| Entry | Yield % | Content of Ni wt.% [a] |
|---|---|---|
| NiBTC–DMF | 56.08 | 31.16 |
| NiBTC–EtOH | 65.63 | 34.64 |
| Ni(pyz)$_2$Cl$_2$ | 50.38 | 19.68 |
| α–Ni(im)$_2$ | 41.22 | 28.47 |

[a] Measured by ICP–OES.

*3.3. Catalytic Test*

Ethylene dimerization was performed in a 200 mL stainless steel autoclave with mechanical stirring, in which 5 bar N$_2$ (three times) and 10 bar ethylene (once) were used to replace air. Then, the autoclave was filled with 70 mL of catalyst solution, including aluminum additive (1 M Et$_2$AlCl in toluene), powdered catalyst, and toluene (solvent). After 30 min, the reaction was stopped by placing it in an ice water bath, and the gas mixture was collected with gas bags. Then, 1 mL of the liquid phase (cooled by ice water and separated from the catalyst by centrifugation) was taken as the sample to be measured by the internal standard method (20 μL n-Heptane is the internal standard). All catalytic data were repeated at least twice, and the error limit of TOF is about ± 4%.

In all cases, the gaseous products were quantified by GC (Techcomp 7890F) equipped with a TM–Al$_2$O$_3$/S column (50 m × 0.53 mm × 25 μm) at 80 °C, direct ramp up to 180 °C. Liquid phase products were quantified by GC (Agilent 6890, Santa Clara, CA, USA) equipped with an HP–5 column (30 m × 0.32 mm × 0.25 μm) at 35 °C for 5 min, heated at 10 °C/min until the temperature achieved 280 °C, which should be maintained 40 min. All of them were coupled with a flame ionization detector (FID).

The activity of the catalyst, the TOF value and the selectivity of the product are calculated from the following equations:

$$\text{Activity} \left( g_{pro.}/(g_{cat.}\cdot h) \right) = m_{all\ products}\ (g) \times m_{cat.}(g)^{-1} \times t\ (h)^{-1}$$

$$\text{TOF} = m_{C_2H_{4Conv.}}(g) \times M_{C_2H_4}^{-1} \times n_{Ni}(mol)^{-1} \times t\ (h)^{-1}$$

$$\left( m_{C_2H_{4Conv.}}(g) = m_{all\ products}\ (g) \right) \quad (1)$$

where in the mass of converted C$_2$H$_4$ ($m_{C_2H_{4Conv.}}$) is equal to the mass of all oligomer products $m_{all\ products}$

$$\text{Selectivity\%} = m_x(g) \times m_{all\ products}\ (g)^{-1} \times 100\%$$

$$m_x\ (x = 1\text{-}C_4,\ \text{Other } C_4,\ C_6 \text{ and } \geq C_8)$$

**4. Conclusions**

Two materials, NiBTC–DMF and NiBTC–EtOH, were synthesized by the one-pot method in two kinds of solvent and characterized by XRD, FT-IR, ICP-OES, XPS, TGA and N$_2$ physical adsorption. After treatment at 150 °C, the catalytic activity of the two materials in ethylene dimerization could reach 470.9 and 647.0 g$_{pro.}$/(g$_{cat.}$·h), and the selectivity of 1-butene in all products could reach 83.2% and 81.7%, respectively, under optimal conditions. Stability testing demonstrated that the catalysts do not decompose during the reaction, but the reusability of catalysts decreases significantly. Furthermore, a probable Cossee–Arlman–type mechanism is proposed. This study reveals the unique potential of CUS-containing MOFs materials in ethylene dimerization.

**Author Contributions:** Data curation, C.W.; investigation, C.W.; methodology, C.W.; supervision, G.L.; validation, C.W. and G.L.; project administration G.L. and H.G.; writing-original draft, C.W.; writing-review and editing, G.L. All authors have read and agreed to the published version of the manuscript.

**Funding:** This research received no external funding.

**Data Availability Statement:** Data are contained within the article.

**Conflicts of Interest:** The authors declare no conflict of interest.

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
