# Peer review of "Ethylene Dimerization Performance of NiBTCs Synthesized Using Different Solvents"

_catalysts, doi:10.3390/catal13030640_

Round 1
Reviewer 1 Report
Reviewer #1:
The manuscript by Guo and co-workers describes the one-pot method for the synthesis of Ni-BTCs in two solvents and characterized by XRD, FT-IR, ICP-OES and XPS. In this study, the catalytic activity of ethylene dimerization and the selectivity of 1-butene in the product were found to be significantly increased for both materials. The authors suggest that this is a result of the direct use of the MOF material as a catalyst. However, there are some points needing further clarifications. It is recommended that the authors give the chemical structure of the MOF material and explain in detail the active centre of the catalytic action. Thereby, I recommend a major revision of the manuscript prior considering the publication of this manuscript in "Catalysts". Below I include some specific comments.
1. In section 3.2 (Catalysts Preparation), it is recommended that the authors give a detailed synthesis schematic so that other researchers who may be interested in the work can repeat fluently.
2. It is recommended that the authors give the specific locations of the characteristic absorption peaks in the FT-IR and make a detailed explanation with a local magnification of 1200 cm-1~1800 cm-1.
3. (1)Page 3 (line 101-102), “861.7eV” corresponding to “Ni 2p 3/2” and “879.7eV” corresponding to “Ni 2p 1/2”; (2)Table 3, “Ni-BTC-DMF-R” in entry 2 should be changed to “Ni-BTC-ETOH-R”.
4. Page 5 (line 170-178), the authors mention that a new characteristic peak appears in XRD, FT-IR of the recovered catalyst. This indicates that the structure of both materials has been changed, resulting in a reduction in their catalytic activity. It is suggested that the author give some methods to improve the reusability of Ni-BTCs.
5. It is suggested that the authors give a thermogravimetric analysis to show that the original structure of Ni-BTC was not destroyed by heat treatment at 150 °C.
6. Table 5, when comparing the performance of ethylene dimerization with published Ni-MOF articles, the data given in the article is too small and the authors are advised to select at least 10 sets of data.
7. (1)Ref. 28 has no DOI number, suggest authors to change reference in order to keep the reference format consistent; (2)Ref. 35, suggest authors to change reference from Co-MOF to Ni-MOF.
8. In Figure S5 (A)and (B), what atoms are represented by each of all the balls should be labelled, and keep both two pictures in the same colour to represent the same atoms, and improve the quality of the uploaded pictures.
9. Page 4 (line 139-141), the authors state that Ni-BTCs instead of metal source or organic ligands possess the catalytic activity in the reaction. However, in the manuscript, the authors do not give details of the active site in Ni-BTCs.
Author Response
Reviewer #1:
The manuscript by Guo and co-workers describes the one-pot method for the synthesis of Ni-BTCs in two solvents and characterized by XRD, FT-IR, ICP-OES and XPS. In this study, the catalytic activity of ethylene dimerization and the selectivity of 1-butene in the product were found to be significantly increased for both materials. The authors suggest that this is a result of the direct use of the MOF material as a catalyst. However, there are some points needing further clarifications. It is recommended that the authors give the chemical structure of the MOF material and explain in detail the active center of the catalytic action. Thereby, I recommend a major revision of the manuscript prior considering the publication of this manuscript in "Catalysts". Below I include some specific comments.
Response:
Thanks for the comment. We have given the probable structure of Ni-BTC (Figure 1) and a description of the generation of the active center at introduction in the manuscript.
1) In section 3.2 (Catalysts Preparation), it is recommended that the authors give a detailed synthesis schematic so that other researchers who may be interested in the work can repeat fluently.
Response:
Thanks, we have given detailed synthesis schematics for each catalyst which are shown in section 3.2 (Scheme 1-3) in the manuscript.
2) It is recommended that the authors give the specific locations of the characteristic absorption peaks in the FT-IR and make a detailed explanation with a local magnification of 1200 cm-1~1800 cm-1.
Response:
Thanks for the helpful suggestion, we have given the specific locations of the catalyst’s characteristic absorption peaks (Figure 2B and 7B) and make a detailed explanation with a local magnification (Figure 2C and 7C) of 1200 cm-1~1800 cm-1 in the manuscript.
3) (1) Page 3 (line 101-102), “861.7eV” corresponding to “Ni 2p 3/2” and “879.7eV” corresponding to “Ni 2p 1/2”; (2) Table 3, “Ni-BTC-DMF-R” in entry 2 should be changed to “Ni-BTC-ETOH-R”
Response:
Thanks, we have modified above two points in the manuscript.
4) Page 5 (line 170-178), the authors mention that a new characteristic peak appears in XRD, FT-IR of the recovered catalyst. This indicates that the structure of both materials has been changed, resulting in a reduction in their catalytic activity. It is suggested that the author give some methods to improve the reusability of Ni-BTCs.
Response:
We thank the reviewer for the meaningful question. Generally, Ni-MOFs materials as ethylene oligomerization catalysts can regain their catalytic activity after being washed using anhydrous ethanol (J. Am. Chem. Soc. 2013, 135, 4195-4198; ACS Appl. Nano Mater. 2019, 2, 136-142) or acidified ethanol (Micropor. Mesopor. Mat. 2022, 338, 111979-111992; Inorg. Chim. Acta. 2023, 544, 121228-121238). We used the first method in the manuscript, which did not work well, so we tried the second method to wash the recovered catalyst by 5% HCl/EtOH solution. However, we found that the newly generated organic components in the structure did not disappear, but the original crystalline shape of the material was severely damaged (below Figure 1).

Figure 1. XRD patterns of Ni-BTC-DMF and Ni-BTC-EtOH.
Therefore, we think this method is not applicable to Ni-BTCs either. We would continue to improve the reusability of Ni-BTCs by reducing the amount of acid or changing the organic solvents (e.g., ether and acetone) in the subsequent study. Considering that the reusability performance of heterogeneous catalysts is not reported in many studies (Organomet. 2017, 36, 632-638; Appl. Cat. A-G 2018, 564, 183-189; ChemCatChem 2019, 12, 135-140; Ind. Eng. Chem. Res. 2022, 61, 14374-14381), we believe that this work is worth to be published.
5) It is suggested that the authors give a thermogravimetric analysis to show that the original structure of Ni-BTC was not destroyed by heat treatment at 150 °C.
Response:
Thanks for the helpful suggestion, we have given a thermogravimetric analysis at Figure 4A and 4B in the manuscript. As shown in these pictures, the initial loss steps of them (Ni-BTC-DMF, 40-178 oC and Ni-BTC-EtOH, 40-229 oC) are attributed to the loss of H2O and some of solvent molecules in coordination in these curves (Angew. Chem. Int. Ed. 2010, 49, 8489-8492; Inorg. Chem. 2011, 50, 5085-5097). Therefore, the original structures of Ni-BTCs are not destroyed by heat treatment at 150 °C.
6) Table 5, when comparing the performance of ethylene dimerization with published Ni-MOF articles, the data given in the article is too small and the authors are advised to select at least 10 sets of data.
Response:
Thanks, we have added 6 sets of comparison data at Table 6 in the manuscript. Now there are 12 sets of data in Table 6.
7) (1) Ref. 28 has no DOI number, suggest authors to change reference in order to keep the reference format consistent; (2) Ref. 35, suggest authors to change reference from Co-MOF to Ni-MOF.
Response:
Thanks for this question, we have removed the two problematic references mentioned above.
8) In Figure S5 (A)and (B), what atoms are represented by each of all the balls should be labelled, and keep both two pictures in the same colour to represent the same atoms, and improve the quality of the uploaded pictures.
Response:
Thanks, we have removed the Supporting Information and presented the structure of the two materials in section 3.2 (Catalysts Preparation Scheme 2-3) with the individual atoms labeled as required. Moreover, we improve the quality of the pictures.
9) Page 4 (line 139-141), the authors state that Ni-BTCs instead of metal source or organic ligands possess the catalytic activity in the reaction. However, in the manuscript, the authors do not give details of the active site in Ni-BTCs.
Response:
Thanks to the reviewer for the suggestion, we have given the probable structure of Ni-BTC (Figure 1) and a description of the generation of the active center at introduction in the manuscript.

Reviewer 2 Report
The manuscript by Wang and co-workers entitled as “Ethylene dimerization performance of Ni-BTCs synthesized using different solvents” reports about the synthesis of Ni-BTCs using one-pot method in two kinds of solvents (DMF and EtOH). Both the materials were found to catalyze the ethylene dimerization reaction with a selectivity of 1-butene (83.2% and 81.7%, respectively). Two other MOFs (α-Ni (Im)2 and Ni(pyz)2Cl2) are also synthesized to compare the catalytic activity with that of Ni-BTCs. The synthesis of Ni-BTC is already well known in literature and the Ni-BTC has been used for OER, supercapacitor, anode material for Li-ion batteries and catalysts for the reaction of CO2 cycloaddition. Although the study is important in terms of selective dimerization of ethylene to 1-butane, however the manuscript should be revised by considering the following comments towards the characterization of Ni-BTC.
1. The PXRD pattern of Ni-BTC-EtOH and Ni-BTC-DMF should be compared with the simulated pattern of the reported Ni-BTCs in order to support the structural characterization of the synthesized materials.
2. Please compare the experimental and simulated PXRD pattern for α-Ni(Im)2 and Ni(pyz)2Cl2.
3. The BET surface areas for Ni-BTC-EtOH and Ni-BTC-DMF should be measured and compared with the existing Ni-BTC in the literature.
4. The activities of both materials were found to decrease after catalysis, Ni-BTC-DMF decreasing to 54% of the original activity and Ni-BTC-EtOH decreasing to 41%. This is not desired. It should be improved.
5. The author showed the XRD patterns of the materials (Figure S1) after the catalysis to prove the retention of structural integrity. However the XRD patterns of the materials before and after catalysis should be plotted together in order to claim the retention of structural integrity of Ni-BTC.
6. Please check the thermogravimetric analysis of Ni-BTCs after activation at 150 ⁰C in order to verify the complete removal of solvent molecule.
Author Response
Reviewer #2:
The manuscript by Wang and co-workers entitled as “Ethylene dimerization performance of Ni-BTCs synthesized using different solvents” reports about the synthesis of Ni-BTCs using one-pot method in two kinds of solvents (DMF and EtOH). Both the materials were found to catalyze the ethylene dimerization reaction with a selectivity of 1-butene (83.2% and 81.7%, respectively). Two other MOFs (α-Ni (Im)2 and Ni(pyz)2Cl2) are also synthesized to compare the catalytic activity with that of Ni-BTCs. The synthesis of Ni-BTC is already well known in literature and the Ni-BTC has been used for OER, supercapacitor, anode material for Li-ion batteries and catalysts for the reaction of CO2 cycloaddition. Although the study is important in terms of selective dimerization of ethylene to 1-butane, however the manuscript should be revised by considering the following comments towards the characterization of Ni-BTC.
Response:
Thanks for the comment.
1) The PXRD pattern of Ni-BTC-EtOH and Ni-BTC-DMF should be compared with the simulated pattern of the reported Ni-BTCs in order to support the structural characterization of the synthesized materials.
Response:
We thank the reviewer for the helpful question. Generally, the CIF file of the material is required to perform the simulations of XRD pattern. However, there are many synthesis methods for Ni-BTC and a variation in synthesis conditions such as Ni source, temperature and reaction time will change its XRD pattern (Chinese Chem. Lett. 2013, 24, 663–667; Chem. Commun. 2014, 50, 9485-9488; J. Colloid. Interface Sci. 2018, 518, 57-68; J. Colloid Interface Sci. 2018, 530, 127-136; J. Ind. Eng. Chem. 2018, 58, 296-303; Int. J. Hydrogen Energ. 2022, 47, 16741-16749). We tried our best to find the standard crystal data that match this work, but unfortunately, we were unable to find it, so we could not perform the simulation. We compare the spectra of Ni-BTCs with those in the previous reports, as shown in below Figure 2. Ni-BTC-DMF showed essentially the same peak positions in the XRD pattern and higher crystallinity compared with Ni-MOF 1-6.

Figure 2. XRD patterns of Ni-BTCs in this work (left) and the Ni-MOF in cited literature (right, J. Colloid. Interface Sci. 2018, 518, 57-68).
2) Please compare the experimental and simulated PXRD pattern for α-Ni(Im)2 and Ni(pyz)2Cl2.
Response:
Thanks for this question, we have added the simulated XRD patterns at Figure 7 in the manuscript.
3) The BET surface areas for Ni-BTC-EtOH and Ni-BTC-DMF should be measured and compared with the existing Ni-BTC in the literature.
Response:
Thanks, we have given the specific surface area by BET measurement of Ni-BTCs at Figure 4 in the manuscript and compared with other Ni-BTCs materials in the literatures at below Table 1. It indicated that the Ni-BTCs synthesized in this work have similar specific surface area as other materials.
Table 1. Comparison of specific surface area of Ni-BTCs in the literatures.
|
Entry |
Time (h) |
Solvent |
specific surface area (m2/g) |
Ref. |
|
Ni-BTCDMF |
24 |
DMF |
198.0 |
J. Colloid. Interface Sci. 2018, 530, 127-136 |
|
Ni(0.5 mmol)-H3BTC-MOF |
24 |
EtOH |
55.78 |
J. Phys. Chem. Sol. 2022, 167, 110743-110768 |
|
Ni-MOF 1-6 |
12 |
DMF |
40.36 |
J. Colloid. Interface Sci. 2018, 518, 57-68 |
|
Ni-BTC-DMF |
12 |
DMF |
60.91 |
This work |
|
NI-BTC-EtOH |
12 |
EtOH |
153.0 |
This work |
4) The activities of both materials were found to decrease after catalysis, Ni-BTC-DMF decreasing to 54% of the original activity and Ni-BTC-EtOH decreasing to 41%. This is not desired. It should be improved.
Response:
We thank the reviewer's comments indeed. Generally, Ni-MOFs materials as ethylene oligomerization catalysts can regain their catalytic activity after being washed using anhydrous ethanol (J. Am. Chem. Soc. 2013, 135, 4195-4198; ACS Appl. Nano Mater. 2019, 2, 136-142) or acidified ethanol (Micropor. Mesopor. Mat. 2022, 338, 111979-111992; Inorg. Chim. Acta. 2023, 544, 121228-121238). We used the first method in the manuscript, which did not work well, so we tried the second method to wash the recovered catalyst by 5% HCl/EtOH solution. However, we found that the newly generated organic components in the structure did not disappear, but the original crystalline shape of the material was severely damaged (below Figure 1).

Figure 1. XRD patterns of Ni-BTC-DMF and Ni-BTC-EtOH.
Therefore, we think this method is not applicable to Ni-BTCs either. We would continue to improve the reusability of Ni-BTCs by reducing the amount of acid or changing the organic solvents (e.g., ether and acetone) in the subsequent study. Considering that the reusability performance of heterogeneous catalysts is not reported in many studies (Organomet. 2017, 36, 632-638; Appl. Cat. A-G 2018, 564, 183-189; ChemCatChem 2019, 12, 135-140; Ind. Eng. Chem. Res. 2022, 61, 14374-14381), we believe that this work is worth to be published.
5) The author showed the XRD patterns of the materials (Figure S1) after the catalysis to prove the retention of structural integrity. However, the XRD patterns of the materials before and after catalysis should be plotted together in order to claim the retention of structural integrity of Ni-BTC.
Response:
We thank the reviewer for the helpful question. We have added the XRD patterns of materials after catalysis at Figure 2 in the manuscript for comparison.
6) Please check the thermogravimetric analysis of Ni-BTCs after activation at 150 ⁰C in order to verify the complete removal of solvent molecule.
Response:
Thanks for the helpful suggestion, we have given a thermogravimetric analysis at Figure 4A and 4B in the manuscript. As shown in the pictures, the initial loss steps of them (Ni-BTC-DMF, 40-178 oC and Ni-BTC-EtOH, 40-229 oC) are attributed to the loss of H2O and some of solvent molecules in coordination in these curves. Therefore, it is true that only part of the solvent molecules is removed at 150 oC. On this point, we will subject Ni-BTC-DMF and Ni-BTC-EtOH to post-activation at 170 oC and 220 oC, respectively, in the follow-up studies.

Reviewer 3 Report
The authors tested the ethylene dimerization activity of a Ni-based MOF. The activity is moderate, but the authors conducted solid systematic works in terms of the evaluation of the catalytic activity. The work has some merit to be published. I think its publication in catalysts is appropriate. The major concern is the stability of the catalyst, as the PXRD pattern changed significantly after the reaction. The authors can perform more analyses of the PXRD pattern to see if any of the NiO or Ni(OH)2 phase formed and can discuss more about the true active form of the catalyst.
Author Response
Reviewer #3:
The authors tested the ethylene dimerization activity of a Ni-based MOF. The activity is moderate, but the authors conducted solid systematic works in terms of the evaluation of the catalytic activity. The work has some merit to be published. I think its publication in catalysts is appropriate. The major concern is the stability of the catalyst, as the PXRD pattern changed significantly after the reaction. The authors can perform more analyses of the PXRD pattern to see if any of the NiO or Ni(OH)2 phase formed and can discuss more about the true active form of the catalyst.
Response:
Thanks for the comment. We have compared the standard XRD spectra of NiO (PDF#14-0177) and Ni(OH)2 (PDF#01-1239) with Ni-BTCs at Figure 2A in the manuscript and did not find the generation of these two phases. So the true active form of the catalyst is not NiO or Ni(OH)2.

Round 2
Reviewer 1 Report
the manuscript has been sufficiently improved to warrant publication in CatalystsAuthor Response
Reviewer #1:
the manuscript has been sufficiently improved to warrant publication in Catalysts.
Response:
Thanks for the comment.
Reviewer 2 Report
The authors revised the manuscript by incorporating the mentioned points. However there are still some missing explanation for structural characterization of Ni-BTC-EtOH and Ni-BTC-DMF. Even the reason for lower BET surface area is not cleared. Although the authors have compared the BET surface area with some literature, Ni-BTC possess higher BET surface area (reference: https://www.sciencedirect.com/science/article/abs/pii/S1350417715301000#:~:text=The%20BET%20isotherms%20of%20Ni,recorded%20for%20the%20MOF%20crystals.)
Please incorporate the mentioned points in the revised manuscript in order to publish this work.
Author Response
Reviewer #2:
The authors revised the manuscript by incorporating the mentioned points. However, there are still some missing explanation for structural characterization of Ni-BTC-EtOH and Ni-BTC-DMF. Even the reason for lower BET surface area is not cleared. Although the authors have compared the BET surface area with some literature, Ni-BTC possess higher BET surface area (reference: https://www.sciencedirect.com/science/article/abs/pii/S13504177153 01000#:~:text=The%20BET%20isotherms%20of%20Ni,recorded%20for%20the%20MOF%20crystals.)
Please incorporate the mentioned points in the revised manuscript in order to publish this work.
Response:
Thanks for the comment. We have given the reason for lower BET surface area in the manuscript. As far as we know, the microporous structure is one of the keys for the high specific surface area of MOFs-like materials. Compared with some other Ni-BTCs with high specific surface area (Ultrason Sonochem. 2016, 31, 93-101. The above-mentioned reference, which is cited in the manuscript as [48].), we found that the microporous area and volume of these materials in this work are approximately equal to zero calculated by the t-plot method and the isotherm shows no absorption at the low specific pressure region (P/P0<0.05), thus indicating that there may not be inherent microporous structures in the materials. This is also the reason for the low specific surface area of these Ni-BTCs. While the low specific surface area does not affect the active centers of these materials and their application. (J. Colloid. Interface Sci. 2018, 518, 57-68; J. Phys. Chem. Sol. 2022, 167, 110743-110751; Appl. Surf. Sci. 2022, 571, 151284-151296).
